# Adjuvant FOLFIRINOX in Patients with Resectable Pancreatic Cancer Is Effective but Rarely Feasible in Real Life: Is Neoadjuvant FOLFIRINOX a Better Option?

**DOI:** 10.3390/cancers15113049

**Published:** 2023-06-03

**Authors:** Yossi Maman, Yaacov Goykhman, Oz Yakir, Alex Barenboim, Ravit Geva, Sharon Peles-Avraham, Ido Wolf, Joseph M. Klausner, Guy Lahat, Nir Lubezky

**Affiliations:** 1Departments of Surgery, Tel-Aviv Medical Center, Sackler School of Medicine, The Nicholas and Elizabeth Cathedra of Experimental Surgery, Tel-Aviv University, Tel-Aviv 69978, Israel; 2Departments of HPB and Transplant Surgery, Tel-Aviv Medical Center, Sackler School of Medicine, The Nicholas and Elizabeth Cathedra of Experimental Surgery, Tel-Aviv University, Tel-Aviv 69978, Israel; 3Institute of Oncology, Tel-Aviv Medical Center, Sackler School of Medicine, The Nicholas and Elizabeth Cathedra of Experimental Surgery, Tel-Aviv University, Tel-Aviv 69978, Israel

**Keywords:** pancreatic cancer, PDAC, FOLFIRINOX, chemotherapy

## Abstract

**Simple Summary:**

In this paper, we showed that only a minority of patients (23%) undergoing upfront pancreatic resection completed the planned 12 courses of FOLFIRINOX, and we delineated the reasons for not completing it. We showed that patients who received neoadjuvant treatment were significantly more likely to receive six treatment courses, and that those who received at least six courses had better overall survival than those who did not. We suggest considering the potential ways of increasing chemotherapy adherence, such as administering treatment before surgery. We believe this article will promote research that has already started—neoadjuvant treatment for patients with resectable pancreatic cancer.

**Abstract:**

Background: The recommended treatment for resectable pancreatic cancer (PC) is resection followed by adjuvant FOLFIRINOX. We assessed the proportion of patients that managed to complete the 12 courses of adjuvant FOLFIRINOX and compared their outcome with that of patients with borderline resectable pancreatic cancer (BRPC) who underwent resection after neoadjuvant FOLFIRINOX. Methods: A retrospective analysis was performed on a prospectively maintained database of all PC patients who underwent resection with (2/2015–12/2021) or without (1/2018–12/2021) neoadjuvant therapy. Results: A total of 100 patients underwent upfront resection, and 51 patients with BRPC received neoadjuvant treatment. Only 46 resection patients started adjuvant FOLFIRINOX, and only 23 completed 12 courses. The main reasons for not starting/completing adjuvant therapy were poor tolerance and rapid recurrence. Significantly more patients in the neoadjuvant group received at least six FOLFIRINOX courses (80.4% vs. 31%, *p* < 0.001). Patients who completed at least 6 courses, either pre- or postoperatively, had better overall survival (*p* = 0.025) than those who did not. In spite of having more advanced disease, the neoadjuvant group had comparable overall survival (*p* = 0.062) regardless of the number of treatment courses. Conclusion: Only a minority of patients (23%) undergoing upfront pancreatic resection completed the planned 12 courses of FOLFIRINOX. Patients who received neoadjuvant treatment were significantly more likely to receive at least six treatment courses. Patients receiving at least six courses had better overall survival than those who received fewer than six courses, regardless of the timing of treatment relative to surgery. Potential ways to increase chemotherapy adherence, such as administering treatment before surgery, should be considered.

## 1. Introduction

Complete margin-negative surgical resection is the only modality offering a chance for cure in patients with pancreatic ductal adenocarcinoma (PDAC). The long-term outcomes of patients undergoing surgery alone, however, are poor, with 5-year survival rates of 10–15% [1]. The crucial role of adjuvant treatment in improving survival outcomes in patients with PDAC is well-documented, and various therapeutic protocols have proven efficacy in improving long-term survival [2,3,4,5]. Conroy et al. [6] demonstrated unprecedented survival for patients receiving adjuvant FOLFIRINOX, with a median survival rate of 54.4 months, which is significantly better than all previously reported ones. However, FOLFIRINOX is a very difficult regimen, with substantial rates of side effects and life-threatening complications. Moreover, the ability of patients to tolerate this regimen after having undergone a major pancreatic operation is not clear. Alternatively, FOLFIRINOX can be given before surgery. The National Comprehensive Cancer Network (NCCN) guidelines recommend pre-surgery neoadjuvant chemotherapy for patients with borderline resectable pancreatic cancer (BRPC) and those with locally advanced PC (LAPC) [7]. Recent studies have demonstrated the safety of this regimen in the preoperative setting [8]. 

Several mostly retrospective studies reported that the long-term oncologic outcomes of patients with BRPC as well as those of selected patients with LAPC undergoing curative resection after neoadjuvant FOLFIRINOX have similar rates of overall survival (OS) compared with those with resectable disease [9,10]. The potential advantages of administering treatment before surgery in this setting are clear. However, there are no level 1 data on the results of administrating neoadjuvant FOLFIRINOX in patients with resectable tumors. 

Our aim was to assess the real-life proportion of patients that underwent curative pancreatic surgery and had managed to complete a full course of adjuvant FOLFIRINOX treatment. We also attempted to delineate reasons for not completing treatment and compared that group with BRPC patients who succeeded in completing neoadjuvant FOLFIRINOX followed by curative surgery. 

## 2. Materials and Methods

### 2.1. Patient Population

The study was approved by the Institutional Review Board of the Tel Aviv Medical Center. Data of all patients who underwent upfront curative pancreatic resection (upfront surgery group) and those who received neoadjuvant FOLFIRINOX for BRPC followed by curative pancreatic resection (neoadjuvant group) in our Department of Surgery between 2015–2021 (the neoadjuvant group) and 2018–2021 (the adjuvant group) were prospectively collected and retrospectively analyzed. 

### 2.2. Initial Evaluation

All patients diagnosed with PDAC underwent a triple-phase contrast-enhanced computerized tomographic (CT) study. Resectability was determined by the pancreatic multidisciplinary team and in accordance with the NCCN guidelines [7]. Patients with resectable tumors were referred for upfront surgery. Those with BRPC/LAPC were first referred for treatment with neoadjuvant FOLFIRINOX to be followed by surgery. All pre-treatment imaging studies were reviewed by a single radiologist (DM) to confirm the classification of tumors as BRPC or LAPC. Chest CT studies and baseline measurements of CEA and CA 19.9 were carried out. A biopsy confirming the diagnosis of PDAC was required for all patients before commencing neoadjuvant therapy.

### 2.3. Chemotherapy 

FOLFIRINOX was administered as reported by Conroy et al. [6]. Response to treatment was evaluated by clinical evaluation, laboratory and imaging findings. Decisions regarding dose/treatment adjustments as well as discontinuation of treatment were made according to oncologic protocols, treatment tolerance and clinical judgment. Surgery was performed within 6 weeks after the last treatment in the neoadjuvant group, and chemotherapy was started up to 12 weeks after the surgery in the upfront surgery group. Progression of disease and deterioration of functional status were an indication to stop treatment and abort plans for curative surgery in the neoadjuvant group. 

### 2.4. Pathologic Characteristics

The histologic analysis was carried out by a single pathologist. Tumor size, presence of peripancreatic fat infiltration, perineural invasion, lymphovascular invasion, surgical margin involvement and lymph node involvement were evaluated and recorded for analysis. 

### 2.5. Follow-Up

OS and progression-free survival (PFS) were determined either from the time of surgery (upfront group) or from the start of neoadjuvant treatment (neoadjuvant group). Follow-up evaluations consisted of abdominal and chest CTs and tumor marker levels, and they were carried out every 3–4 months in the first year, every 6 months in the second year, and every year thereafter. A total of 134 (89%) patients completed the planned follow-up.

### 2.6. Statistical Analysis

Categorical variables were described by incidences and percentages and compared by a Chi-square test. Quantitative variables were described by averages and standard deviations or by medians and compared with an independent *T*-test and the Mann–Whitney U test. The survival analysis was calculated with the Kaplan–Meier method. The statistical analysis was performed with SPSS version 26.

## 3. Results

### 3.1. Patient Demographics and Clinical Characteristics (Table 1)

The upfront surgery group included 125 patients of whom 25 patients above the age of 79 years were excluded (non-candidates for FOLFIRINOX treatment according to the NEJM trial [6]). The neoadjuvant group included 51 patients with BRPC. Clinically, the neoadjuvant group was characterized by more signs and symptoms of advanced disease, including back pain (*p* = 0.007), anorexia (*p* = 0.001) and higher CEA levels (*p* = 0.001). 

**Table 1 cancers-15-03049-t001:** Demographic and clinical characteristics of the patients.

Variable	Upfront Surgery	Neoadjuvant Chemotherapy	*p*-Value
	*n* = 100 *n* (%)/Average (SD)	*n* = 51 *n* (%)/Average (SD)	
Sex, male	51 (51)	29 (56.9)	0.495
Age, years	66.4 (9.8)	63.2 (8.5)	0.963
Body mass index	25.6 (4.3)	24.4 (3.7)	0.287
Smoker	21 (21)	8 (16.3)	0.498
Charlson Comorbidity Index	5.1 (1.5)	4.4 (1.2)	0.103
Back pain at presentation	17 (17)	19 (52.8)	**0.007 ^1^**
Weight loss > 10%	32 (32)	30 (68.2)	0.153
Loss of appetite	34 (34)	22 (66.7)	**0.001**
Duration of symptoms, days	78 (135)	123 (144)	0.165
Biliary drainage, ERCP/PTD ^2^	34 (34)	17 (33.3)	0.935
CA19-9, mg/dL ^3^	643 (3305)	700 (1262)	0.883
CEA ^3^	2.3 (2.0)	6.3 (9.9)	**0.001**
Bilirubin, maximum	7.4 (7.0)	4.1 (5.9)	**0.007**
Sex, male	51 (51)	29 (56.9)	0.495
Age, years	66.4 (9.8)	63.2 (8.5)	0.963

^1^ Bold indicated significant; ^2^ ERCP/PTD endoscopy retrograde cholangiopancreatography, percutaneous transhepatic drainage; ^3^ Levels before surgery.

### 3.2. Operative and Pathological Characteristics (Table 2)

Patients in the neoadjuvant group had longer duration surgeries (*p* = 0.001) and higher rates of vascular resection (*p* < 0.001) compared to the upfront surgery group. The final pathology showed a lower rate of nodal involvement (*p* = 0.003). The R0 resection rate was not different between the two groups (*p* = 0.823). 

**Table 2 cancers-15-03049-t002:** Operative and pathologic characteristics.

Variable					Upfront Surgery	Neoadjuvant Chemotherapy	*p*-Value
					*n* = 100 *n* (%)/Average (SD)	*n* = 51 *n* (%)/Average (SD)	
**Operative parameters**							
Surgery Type	Whipple				81 (81)	26 (51)	**<0.001**
	Distal/subtotal pancreatectomy + splenectomy				15 (15)	18 (35.3)	
	Total pancreatectomy				3 (4)	7 (13.7)	
Length of surgery, minutes					408 (106)	476 (149)	**0.001**
Vascular resection ^1^					22 (22)	32 (62.7)	**<0.001**
Intraoperative mortality					0	0	-
**Postoperative course**							
LOS ^2^ after surgery (days)					16.4 (14)	17.4 (16)	0.600
ICU ^3^ admission					78 (78)	46 (90.2)	0.064
Time in ICU (days)					2.6 (3.9)	3.9 (5)	0.486
Complications					67 (67)	29 (56.9)	0.221
Clavien-Dindo classification				0–2	87 (87)	41 (80.4)	0.285
				3–5	13 (13)	10 (19.6)	
Need for revision surgery					6 (6)	4 (8.5)	0.573
**Pathology**							
			Well diff. ^4^		12 (12)	9 (21.4)	0.207
Grade			Mod. diff. ^4^		76 (76)	28 (66.7)	
			Poorly diff. ^4^		6 (6)	5 (11.9)	
Tumor greatest dimensions					2.9 (1)	2.9 (1.5)	0.091
Affected nodes (N+ disease)					51 (51)	13 (25)	**0.003**
Total number of nodes					24 (11)	20.8 (11.8)	0.756
**R0 resection**					**95 (95)**	**46 (96)**	**0.823**
PNI ^5^					44 (44)	23 (51.1)	0.523
LVI ^6^					23 (23)	10 (22.7)	0.898
Pathology staging		Stage 0			0 (0)	2 (5.6)	**<0.001**
		Stage IA			11 (11)	12 (33.3)	
		Stage IB			33 (33)	8 (22.2)	
		Stage IIA			3 (24.6)	4 (11.1)	
		Stage IIB			36 (36)	10 (27.8)	
		Stage III ^7^			14 (14)	0 (0)	

^1^ Routine resection of splenic vein/artery in distal/total pancreatectomy was not considered as “vascular resection”; ^2^ LOS—length of stay, ^3^ ICU—intensive care unit, ^4^ diff—differentiated; ^5^ PNI—perineural invasion; ^6^ LVI—lymphovascular invasion; ^7^ None of the 14 patients with stage III pancreatic cancer had T4 disease, and all 14 were assigned stage III based upon N2 disease. Bold indicates significant.

### 3.3. Treatment

Only 23 (23%) of the patients in the upfront surgery group managed to complete 12 courses of FOLFIRINOX compared to 13 patients (25.5%) in the neoadjuvant group (*p* = 0.608). The main reasons for not completing the adjuvant course were patient refusal and rapid recurrence in the upfront surgery group and prolonged recovery in the neoadjuvant group (Figure 1). Nevertheless, more patients in the neoadjuvant group received at least six courses of FOLFIRINOX (80.4% vs. 31%, *p* < 0.001). It is important to note that patients in the neoadjuvant group also received adjuvant FOLFIRINOX treatment. On average, they received 6.3 neoadjuvant treatments (SD 2.7) and 4.7 adjuvant treatments (SD 3). Additionally, 12 out of 47 patients received more than six treatments postoperatively. The attrition rate for patients in the neoadjuvant course was 13% for those with BRPC [11].

### 3.4. Survival

Patients who completed 12 courses of FOLFIRINOX regardless of the timing of chemotherapy had significantly better OS and PFS then those who did not. The median OS and PFS for those who completed 12 courses did not reach 50% compared with a median OS and PFS of 29 months and 16 months, respectively (*p* = 0.046 and *p* = 0.006, respectively), for those who did not.

The survival benefit was also significant for patients who received at least 6 courses of FOLFIFIRNOX compared to those who received fewer than 6 courses (*p* = 0.025) (Figure 2). 

## 4. Discussion

The survival of patients with pancreatic cancer is notoriously poor. Most studies demonstrated that even patients undergoing curative resection have an expected median survival of only 15–20 months [1]. Recent studies have described the crucial role of adjuvant treatment in patients with PDAC who undergo curative resection. Adjuvant gemcitabine and, later, a combination of gemcitabine with capecitabine resulted in improved outcomes, with median survival rates of about 30–35 months [3,4,5]. The trial by Conroy et al. [6] published in the New England Journal of Medicine in 2018 demonstrated the unprecedented impact of adjuvant modified FOLFIRINOX on long-term survival in patients undergoing curative resection for pancreatic cancer. Compared to gemcitabine, adjuvant treatment with modified FOLFIRINOX resulted in a median survival of 54.4 months compared with 35 months in the gemcitabine group (*p* < 0.001), making modified FOLFIRINOX the adjuvant treatment of choice in these patients. 

Unfortunately, FOLFIRINOX is a toxic regimen that is very difficult to tolerate, especially among patients that had recently undergone a major pancreatic resection. In Conroy et al.’s study, 66.4% of the patients received all of the planned cycles of chemotherapy. In our study, only 46% of the patients started adjuvant FOLFIRINOX, and only 23% managed to receive all 12 cycles. Moreover, as expected, our data demonstrated that the median survival in patients who failed to complete 12 courses of FOLFIRINOX was significantly shorter than that of patients who succeeded (*p* = 0.004). The main reasons for not undergoing adjuvant treatment or not completing the fully planned 12 courses of FOLFIRINOX were patient refusal, rapid cancer recurrence and poor tolerance of treatment. More patient counselling on the importance of adjuvant FOLFIRINOX in determining long-term outcome may be required. Minimal invasive surgery and enhanced recovery from surgery could also contribute to an increased rate of treatment adherence. Administering part or all of the 12 courses of treatment before surgery is another option to be considered. There is an ongoing randomized controlled trial evaluating the efficacy of neoadjuvant compared with adjuvant FOLFIRINOX [12], and this trial will hopefully clarify this issue. 

The currently recommended treatment for patients with BRPC is neoadjuvant treatment with FOLFIRINOX followed by curative resection whenever feasible. Although there are currently no data from prospective randomized trials, many observational studies have demonstrated the safety and efficacy of neoadjuvant FOLFIRINOX in these patients. Patients operated after neoadjuvant therapy are reported to have similar perioperative mortality rates as those who underwent upfront surgery [8,13,14], and there are reports that perioperative morbidity in patients operated after neoadjuvant therapy is lower due to lower rates of pancreatic fistula resulting from the hard texture of the pancreas in treated patients [14,15]. Most studies report high margin-negative resection rates and long-term survival that is comparable to that in patients with resectable pancreatic cancer [11,16]. A comparison of our data of patients with BRPC who were operated after neoadjuvant FORFIRINOX to those of patients with resectable tumors undergoing upfront resection revealed that significantly more patients in the neoadjuvant group managed to receive at least six courses of FOLFIRINOX (80.4% versus 34%, respectively, *p* < 0.001), and that patients who completed at least six treatment courses had a significantly better OS (*p* = 0.013) compared with patients who received fewer than six courses of FOLFIRINOX, regardless of the timing of treatment relative to surgery. Moreover, the long-term outcomes of patients in the neoadjuvant group were not worse in spite of their having more advanced local disease regardless of the amount of treatment courses. 

Although only a prospective randomized trial can determine whether there are advantages to neoadjuvant over adjuvant treatment, there are several other potential advantages of neoadjuvant treatment, such as downsizing the tumor, improving R0 resection rates, early administration of systemic treatment to micro-metastatic disease and administering treatment to more patients. The main concerns with neoadjuvant therapy are progression of disease during the treatment and increased morbidity among patients undergoing surgery after treatment. The results of our current study demonstrate that there was no increase in perioperative morbidity among those who underwent pancreatic surgery after treatment. Our data also showed that patients with resectable PDAC undergoing resection followed by adjuvant treatment and patients with more advanced BRPC undergoing resection after neoadjuvant FOLFIRINOX have similar long-term survival rates, a finding that could support the shift of the paradigm in favor of the administration of neoadjuvant FOLFIRINOX in patients with resectable pancreatic cancer. 

This study has a number of limitations. First, this was a retrospective analysis of a prospectively maintained database and, therefore, subject to bias. One of the major limitations of this study is the differentiation between the cohorts. The comparison of patients operated after neoadjuvant treatment to those in the upfront surgery group is clearly biased, as those receiving neoadjuvant treatment have more advanced disease and are deemed fit for neoadjuvant FOLFIRINOX. Moreover, the number of patients that started neoadjuvant therapy and did not undergo curative surgery was not analyzed in this study. Additionally, our surgical division is considered a referral center in Israel, and many of the operated patients received neoadjuvant treatment at other centers. Therefore, our analysis of attrition rates is limited to patients receiving treatment at the Tel-Aviv Medical Center.

## 5. Conclusions

In conclusion, our data demonstrated that only 23% of patients undergoing upfront resection of resectable pancreatic cancer who were eligible for adjuvant FOLFIRINOX completed the intended treatment plan of 12 courses of FOLFIRINOX, and that these patients had better OS compared with patients that did not undergo treatment. Patients who received neoadjuvant treatment were significantly more likely to receive at least six treatment courses. Patients receiving at least six courses had better overall survival than those who received fewer than six courses, regardless of the timing of treatment relative to surgery. When considering the crucial role of chemotherapy in determining long-term survival of patients with potentially curative pancreatic cancer, methods to increase patient adherence to treatment, such as administering treatment before surgery, need to be considered.

## Figures and Tables

**Figure 1 cancers-15-03049-f001:**
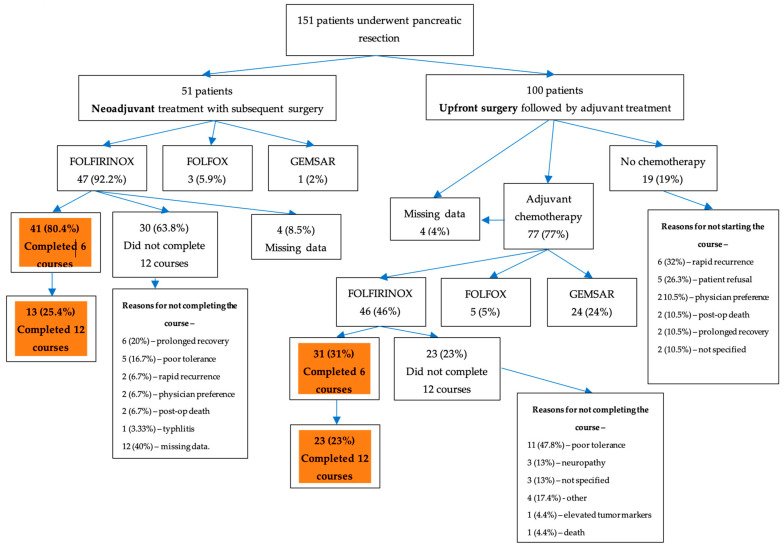
Treatment distribution.

**Figure 2 cancers-15-03049-f002:**
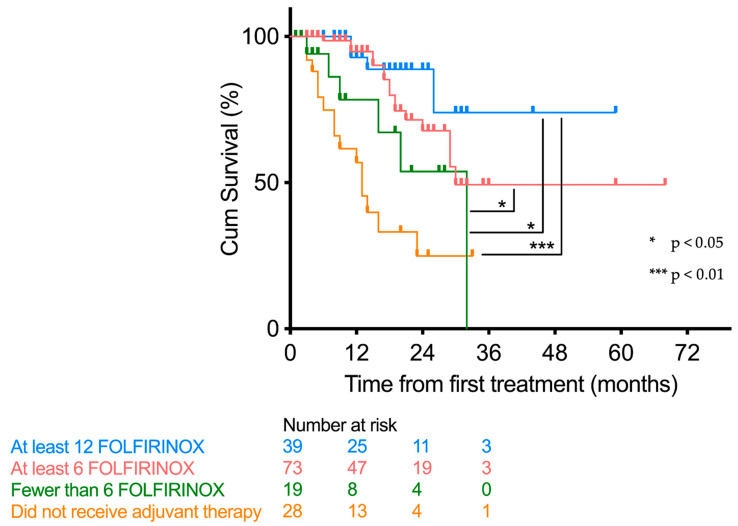
Survival plots divided by number of treatment courses received.

## Data Availability

Data is available on request due to privacy/ethical restrictions.

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
