# Peer review of "Adjuvant FOLFIRINOX in Patients with Resectable Pancreatic Cancer Is Effective but Rarely Feasible in Real Life: Is Neoadjuvant FOLFIRINOX a Better Option?"

_cancers, 2023, doi:10.3390/cancers15113049_

Round 1

Reviewer 1 Report (Previous Reviewer 1)

The authors addressed concerns raised previously.

Author Response

Thank you for your guidance throughout this process. we sincerely appreciate your valuable comments.  

Reviewer 2 Report (Previous Reviewer 2)

The authors well responded to reviewers' comments. I would like the authors to present in the Fig. 2 the numbers of patients in each of the four groups and in which comparison the difference was statistically significant.

Author Response

thank you very much for your guidance throughout the process. we sincerely appreciate your valuable feedback. 

we revised Figure 2  and added the number of patients in each group and specified the comparisons in which statistically significant differences were observed. 

the new figure is attached to the cover letter. 

This manuscript is a resubmission of an earlier submission. The following is a list of the peer review reports and author responses from that submission.

Round 1

Reviewer 1 Report

In this study, cohorts were very dispersed and comparisons were made between cohorts of very different disease settings, which is hard to know its relevance.

Results in Figure 2 may mislead readers: neoadjuvant therapy did not give significant benefit. However, cohorts compared in this Figure 2 seemed to be very different. The neoadjuvant group consisted of BRPCs; while the upfront surgery group consisted of resectable PCs. Figure 2 should be deleted because it does not make sense.

In Figure 3, patients treated with neoadjuvant and adjuvant were mixed. Since cohorts of neoadjuvant and adjuvant were very different, and relevance of such mixture is very hard to understand.

Differences between studies in Figure 3 and Figure 4 are not clear.

In Figure 5, patients completed 6 courses of FOLFIRINOX showed longer median survivals than those completed 12 courses. Does this mean 6 courses were better?

In Figure 6, cohorts of neoadjuvant and those of adjuvant were very different. The comparison between such different cohorts was hard to know its relevance.

Reviewer 2 Report

This is a clinically interesting paper showing surprisingly low completion rate of postoperative adjuvant FOLFIRINOX. The results were well presented, but there are several concerns to be addressed.

1. The description that at least 6 courses of FOLFIRINOX "regardless of the timing relative to surgery" seems to suggest the patients should undergo at least 6 courses pre- and post-operatively in total. The authors should clearly explain whether the patients with neoadjuvant chemotherapy received postoperative adjuvant FOLFIRINOX. If so, how many patients received at least 6 or 12 courses only after postoperative adjuvant treatment should be presented.

2. The authors mentioned in the discussion that the the OS of patients in the neoadjuvant group who received the full 12 courses of FOLFIRINOX was similar as that of those in the adjuvant group who received 12 courses. However, this comparison was not presented in the results.

3. Please present how many of the 14 patients with stage III in upfront surgery group had T4 disease.

4. Please also present R0 resection rate in each of upfront surgery and neoadjuvant chemotherapy group.

5. Figure legends should be presented especially for figure 2-6. In addition, the figures should be modified to fit for the presentation as a manuscript rather than just the copy of the results in SPSS software.

6. I wonder how DFS from the start of neoadjuvant chemotherapy because patients were not disease free before surgical resection.